# Thermal Modeling of the GaN-based Gunn Diode at Terahertz Frequencies

**Ying Wang** [1] **, Jinping Ao** [2,*]**, Shibin Liu** [1] **and Yue Hao** [2]

[1] School of Electronic Information, Northwestern Polytechnical University, Xi'an 710072, China; yingwang@nwpu.edu.cn (Y.W.); liushibin@nwpu.edu.cn (S.L.)
[2] The State Key Discipline Laboratory of Wide Band Gap Semiconductor Technology, School of Microelectronics, Xidian University, Xi'an 710071, China; yhao@xidian.edu.cn
* Correspondence: jpao@ee.tokushima-u.ac.jp; Tel.: +86-029-88491212



**Featured Application: This manuscript provides a simulation study on the self-heating effect of the GaN-based terahertz Gunn diode. It would potentially give some guidance to the design of the Gunn diode and other terahertz oscillators. The method in this manuscript can be also used to study the self-heating effect of other semiconductor devices.**

**Abstract:** In this paper, a comprehensive evaluation of thermal behavior of the GaN vertical $n^+$-$n^-$-$n$-$n^+$ Gunn diode have been carried out through simulation method. We explore the complex effects of various parameters on the device thermal performance through a microscopic analysis of electron movements. These parameters include operation bias, doping level, and length of the active region. The increase of these parameters aggravates the self-heating effect and degrades the electron domains, which therefore reduces the overall performance output of the diode. However, appropriate increase of the doping level of active region makes the lattice heat distribute more uniformly and improves the device performance. For the first time, we propose the transition domain, which is in between the dipole domain and accumulation layer, and stands for the degradation of the electron domain. We have also demonstrated that dual domains occur in the device with longer active region length and higher doping level under EB (Energy balance) model, which enhances the harmonics component. Electric and thermal behaviors analysis of GaN vertical Gunn diode makes it possible to optimize the device.

**Keywords:** GaN Gunn diode; terahertz oscillator; thermal modelling; self-heating effect; III–V semiconductor devices

## 1. Introduction

As one of the excellent candidates for terahertz radiation source, Gunn diodes have attracted much attention [1–8]. Many recent researches demonstrate that the negative-differential-resistance (NDR) based GaN devices possess outstanding power performance at terahertz frequencies due to its unique electronic properties such as large band gap, high breakdown electric field, high electron drift velocity, and large NDR threshold voltage. Although theoretical works predict Gunn oscillations in the THz range in GaN diode [4–7], experimental demonstrations of oscillations have not been achieved due to the technological bottleneck on GaN epitaxial growth and device fabrication. Besides, GaN material would face a serious self-heating problem when it is used for NDR device, because the threshold electric-field for NDR effect is almost 50 times larger than the value of GaAs material. For the Gunn diode, self-heating may trigger the weakening of the NDR effect and even the permanent failure of diode [8–11]. As far as we know, there are many theoretical studies on Gunn diode recent

years, but without considering the self-heating effect. Some conclusions derived by these studies may not stand when taking the self-heating effect into consideration, particularly when the operation frequency approaches the terahertz regime. It is imperative to thoroughly investigate the relation between the thermal and electrical properties of terahertz Gunn diode in order to optimize the structure of the GaN Gunn diode and prevent or minimize the joule heating. There are great limitations to study the device thermal performance on the experimental method. Firstly, as we mention above, the fabrication and measurement of GaN Gunn diode are still very difficult because of bottlenecks on GaN epitaxial growth and device fabrication and the serious self-heating problem. Secondly, the self-heating effect deteriorates the output characteristics of the Gunn diode by affecting the motion of the electrons. However, it is difficult to provide a microscopic analysis on electron movements (especially the electron domain movements) by experimental method. Therefore, we choose the physical-based modeling and simulation method to achieve accurate device thermal management, which would guide the experiment research in turn.

It is of difficulty to clarify the thermal effect on the electron domain in such a short transit region of Gunn diode at such a high electric field by means of traditional device transport theory. In this work, we use the energy balance transport model to explore the self-heating effect on characteristics of GaN vertical Gunn diode by means of a 2-D device simulation platform. The papers published on Gunn diode recent years have been mainly concentrated on the lateral high electron mobility transistor (HEMT)-like Gunn diode, due to its advantages over vertical ones [12–14]. However, the lateral structure still face reliability and integration challenges. Compared with the lateral diode, the vertical one also has its advantages, such as easier package, higher breakdown voltage, and so on [15–17]. Furthermore, the dc-to-ac dispersion induced by the surface states in lateral diode would be mitigated in the vertical one. Therefore, study on vertical Gunn diode is also needed. In this paper, we analyze the effect of the operation bias, the doping level and length of the active region on the performance and heating generation of the diode in detail. The geometry of the vertical diodes and physical model are described in Section 2, results and discussions are given in Section 3, and conclusions are given in Section 4. In order to better understand the thermal and electrical properties of GaN Gunn diode, two transport models, the non-isothermal-energy-balance model (NEB) and the energy-balance model (EB) are compared alternately in the simulation of each device sample.

## 2. Method of Analysis and Physical Models

Compared with the drift-diffusion (DD) model, the EB model is more accurate for spatiotemporal domain and chaos in the short channel. This is because the EB model includes non-localized carrier transport phenomena, such as the velocity overshoot and the non-local impact ionization which are neglected by the conventional DD model. Therefore, the EB model is more suitable for the transient simulation of terahertz devices and is employed in this paper. Indeed, the non-local transport phenomena can be modeled by using Monte Carlo method or using higher order moments of the Boltzmann transport equation, though huge computation times are required by these approaches. In this paper, higher order solution to the general Boltzman transport equation is solved based on the EB model at the Atlas simulation platform, which consists of an additional coupling of the current density to the carrier' temperature, or energy [18]. MODELS statement is the important part in the device-simulation code, where the physical models of the device is specified. In the simulation code of Gunn diode in Atlas, the energy balance model is selected by specifying the 'HCTE' command in the MODELS statement to enable the energy transport model. The energy relaxation time $\tau_\varepsilon$ is defined as 150 fs based on [4,19]. Meanwhile, the self-heating simulations are incorporated in the GaN Gunn diode structures using GIGA module of Atlas to account for lattice heat flow and general thermal environments. GIGA module accounts for the dependence of material and transport parameters on the lattice temperature. For simulation of self-heating effects, it adds the heat flow equation to the primary equations that are solved by Atlas. The heat flow equation has the following expression:

$$C\frac{\partial T_L}{\partial t} = \nabla(\kappa \nabla T_L) + H \tag{1}$$

where C is the heat capacitance per unit volume, $\kappa$ is the thermal conductivity, H is the heat generation, and $T_L$ is the local lattice temperature.

In our simulation, the thermal conductivity is set as a temperature-dependent thermal conductivity: $\kappa = \kappa_{300}/(T/300)^{\alpha}$, where $\kappa_{300}$ is the thermal conductivity at the temperature of 300 K and $\alpha$ is the temperature dependence coefficient [20]. For GaN, $\kappa_{300}$ is set equal to 1.6 W/cm·K and $\alpha$ = 1.4 [20]. The heat capacitances C are 3.0135 J/cm$^3$/K for GaN. All the interfaces between the diode and external surroundings are set to be adiabatic boundary conditions. In order to simplify the algorithm and solve the non-convergence problem, the heat transport from the metal contact to external surroundings is not included in our simulations, a common practice in the simulation of thermal effect as described in [21–23]. The LAT.TEMP lattice temperature model is added to the MODELS statement to include heat flow. We have used a non-isothermal energy balance (NEB) model by specifying both HCTE. and LAT.TEMP parameters in the MODELS statement to solve both energy balance equations and heat flow equations. The NEB model is an extension of stratton's energy balance model for the case of nonuniform lattice temperature, which is a set of six partial differential equations for electrostatic potential, electron and hole concentrations, electron and hole carrier tempertures, and lattice temperature [18]. For the impact ionization model, we adopt Toyabe impact ionization model, which is available when the energy balance transport model is applied [18,24]. A key parameter of the model is the energy balance length $L_{REL}$, which can be calculated by $L_{REL} = \upsilon_{sat} \times \tau_\varepsilon$, where $\upsilon_{sat}$ are the saturation velocities for electrons, and $\tau_\varepsilon$ corresponds to the energy relaxation time.

In this paper, we put an emphasis on n$^+$-n$^-$-n-n$^+$ structural GaN Gunn didoes, where a notch-doped n$^-$ region adjacent to the cathode of diode aims to create a high electric field zone so as to promote the onset of electron domain in the cathode. Figure 1 gives the schematic of the strucuture of the diode we study. As described in Figure 1, we propose a fixed length ($L_{notch}$) of 250 nm and a fixed doping concentration ($N_{notch}$) of $5 \times 10^{16}$ cm$^{-3}$ for the notched layer. The highly doped n$^+$ ohmic contact region (n$^+$ layer) have a fixed length of 100 nm. The length of transit region ($L_{ac}$) ranges from 0.8 μm to 1.4 μm, and its doping concentration ($N_{ac}$) ranges from $1.5 \times 10^{17}$ cm$^{-3}$ to $4 \times 10^{17}$ cm$^{-3}$. Without speciously speaking, $L_{ac}$ = 0.8 μm, $N_{ac}$ = $1.5 \times 10^{17}$ cm$^{-3}$, and the operation bias voltage $V_{dc}$ = 50 V. In the simulation, in order to sustain an average electric filed in the transit region of Gunn diode, we intentionally change the value of $V_{dc}$ in proposition to $L_{ac}$, for instance, if $L_{ac}$ = 1.0μm, then $V_{dc}$ = 62.5 V, and if $L_{ac}$ = 1.4 μm, then $V_{dc}$ = 87.5 V. In order to improve the calculation efficiency of the numerical simulation, we put a single-tone sinusoidal voltage of form $V_{dc} + V_{ac}\sin(2\pi ft)$ across the diode instead of embedding it to an RLC resonant circuit to calculate the RF (Radio Frequency) output power $P_{RF}$ and the dc-to-ac conversion efficiency η of the diodes, as the external circuit adds complexity of the calculation and easily leads to the non-convergence problems [25].

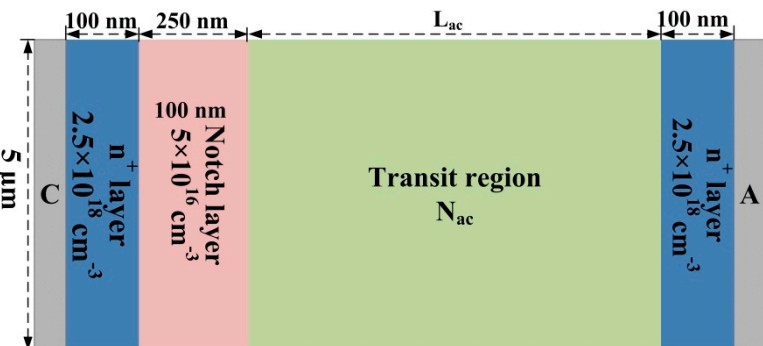

**Figure 1.** Schematic structure of n$^+$-n$^-$-n-n$^+$ structural GaN Gunn didoe.

## 3. Result and Discussion

Figure 2 gives direct current I–V characteristics of the GaN Gunn diode under the EB model and NEB model, respectively. A serious drop of the saturate current appears when taking the self-heating problem into consideration in NEB model, which is attributed to the decrease of the electron velocity in the transit region due to the increased lattice temperature. At smaller $V_{dc}$, however, the differences in the anode current are almost negligible, because the increment of the lattice temperature is small. The heat generation in the transit region is caused by an energy transferring from the electron to the lattice, which is related to inelastic phonon scattering processes of the electron. As reported in [26], the scattering processes mainly responsible for the heat generation are the intra-valley scatterings of the electron in the Γ1 and the U valley. The heat generation originated from the intra-valley scattering in the Γ1 valley builds up near the cathode side. The electric field peak at the notch layer increases the scattering rate and accelerates the electrons to attain sufficient energy to transfer to the upper U-valley. Therefore, as the electrons travel towards the anode side, the heat generation originated from the intra-valley scattering in the U and other upper valleys begins to predominate. The variation of lattice temperature throughout the transit region can be investigated by using NEB model. Our simulations shown that the peak lattice temperature occurs near the notch-doped region where the electric field has a maximum, approaching 820 K at $V_{dc}$ = 50 V and 1015 K at $V_{dc}$ = 60 V, respectively. Therefore, we suggest using a pulsed bias operation to reduce the self-heating generation, in consistent with the fact that short bias pulses are normally applied instead of dc biases in order to avoid possible joule in the experiments of GaAs Gunn diode and HEMT devices [15,27–30].

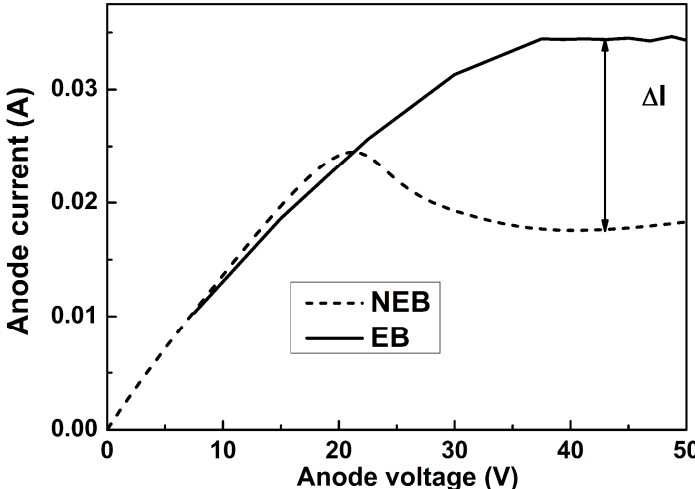

**Figure 2.** Simulated direct I–V characteristics of GaN Gunn diode under NEB (non-isothermal-energy-balance model) model and EB (energy-balance model) model.

In the instantaneous simulation, stable oscillations generate when the bias voltage of GaN Gunn diode ranges from 50 V to 75 V under EB model. However, stable oscillations can only be obtained at a narrow voltage range of 50–55 V under NEB model, which attributes to influence of thermal generation. For both models, the dc-to-ac conversion efficiency η decreases with $V_{dc}$, as shown in Figure 3, which gives the results of η, $f_{osc}$ (oscillation frequency) and $P_{RF}$ versus $V_{dc}$. Under NEB model, as the operation voltage rises from 50 to 55 V, there is an obvious increase of lattice temperature throughout the whole diode, especially near the anode side, as manifested in Figure 4. The increase of the electric field greatly enhances the phonon scatterings of the electrons, which leads to the increase of the lattice temperature with $V_{dc}$, and accordingly the subsequent decrease of the anode current and the conversion efficiency. The serious heat accumulation even prevents the formation of stable oscillation when the bias voltage becomes higher than 55 V. At the bias voltage of 50 V, we extract the oscillation frequency $f_{osc}$ of around 207.0 GHz and 178.1 GHz from stable oscillation currents

under EB and NEB model, respectively. We can also find from Figure 3 that $f_{osc}$ shows an uptrend under both models as the voltage increases, which can be explained by Figure 5, the electric field profiles derived inside one oscillation period. At the operation voltage of 50 V, diode operates at dipole domain mode under EB model, as manifested in Figure 5a. Figure 5b shows that under EB model, the electron domain degrades into the mode between dipole domain and accumulation domain at $V_{dc} = 75$ V, and we call it a transition domain mode for the first time. As compared Figures 4c and 5d, we conclude that the joule heating generation accelerates the degradation of the domain under NEB model. Only transition domain formed under NEB model as $V_{dc} = 50$ V. When the operation voltage rises to 55 V, the domain completely degrades to accumulation domain. In theory, the dipole-domain mode oscillation is more stable and generates higher RF power compared with other oscillation modes in the Gunn diodes. As a result, no matter under EB model or NEB model, the RF output power $P_{RF}$ shows a downward trend, although the voltage increases, as shown in Figure 3b The dipole domain is composed of an excess electrons layer and a depleted electrons layer. However, there is only an excess electrons layer in the accumulation layer. Therefore, dipole-domain mode generates a lower frequency than the accumulation-layer mode does. As a conclusion, the degradation of the domain under both EB and NEB model incurs a decrease in conversion efficiency and RF output power and a slight increase in frequency.

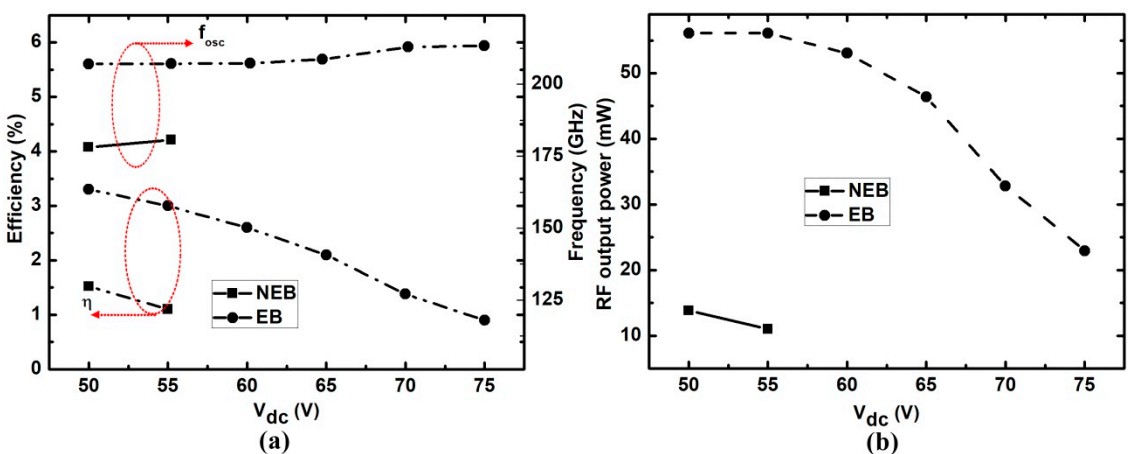

**Figure 3.** (**a**) Simulation results of the oscillation frequency $f_{osc}$ and dc-to-ac conversion efficiency η versus the anode voltage $V_{dc}$ and (**b**) RF output power $P_{RF}$ versus $V_{dc}$ under EB and NEB models.

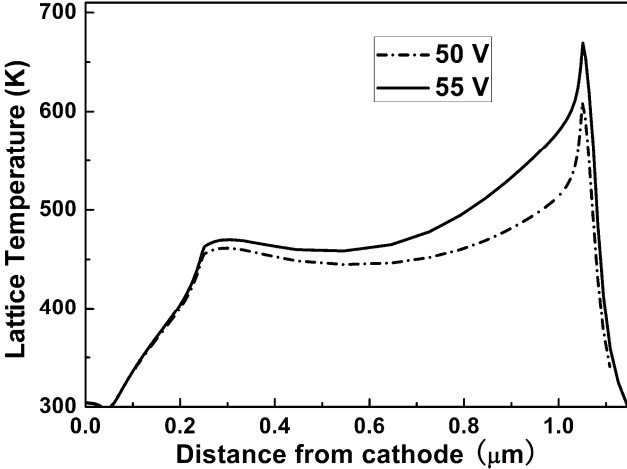

**Figure 4.** Lattice temperature profiles at different bias voltages under NEB model.

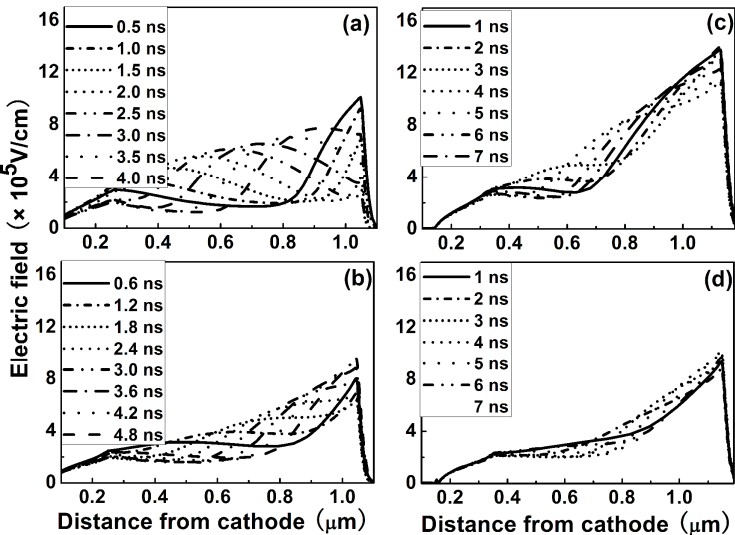

**Figure 5.** Electric field profiles derived inside one oscillation circle at the same time steps, when the diode is applied at different voltage: (**a**) $V_{dc}$ = 50 V and (**b**) $V_{dc}$ = 75 V under EB model; and (**c**) $V_{dc}$ = 50 V and (**d**) $V_{dc}$ = 55 V under NEB model.

The influence of different doping concentration of transit region $N_{ac}$ on the output characteristics and physical mechanism of GaN Gunn diode is also investigated. We get the direct I–V and the global temperature characteristics for $N_{ac}$ ranging from $1.5 \times 10^{17}$ to $4 \times 10^{17}$ cm$^{-3}$ under EB model and NEB model, as shown in Figure 6. Under EB model, the saturated current characterizing the generation of NDR oscillation exhibits a nearly uniform increase with a rising $N_{ac}$. However the increase of the saturated current obtained under NEB model becomes smaller. At higher doping levels, anode current cannot become saturated as the voltage increases under NEB model, particularly, it presents obvious upward trend when $N_{ac}$ is larger than $2 \times 10^{17}$ cm$^{-3}$. This is because the increase of the lattice temperature with $N_{ac}$ degrades the negative resistance effect. In Figure 6, the global lattice temperature can approach 1080 K when $N_{ac} = 2.5 \times 10^{17}$ cm$^{-3}$. Our simulation also presents the upper limit doping level of $N_{ac}$ for stable oscillations is $3 \times 10^{17}$ cm$^{-3}$ under EB model and $2.5 \times 10^{17}$ cm$^{-3}$ under NEB model. Figure 7 shows the influence of $N_{ac}$ on dc-to-ac conversion efficiency, oscillation frequency and RF output power. Under EB model, η decreases as doping level increases, however, there is an obvious increase of η under NEB model at $N_{ac}$ of $2 \times 10^{17}$ cm$^{-3}$. The same trend also appears on the curves of $P_{RF}$-$N_{ac}$ as shown in Figure 7b. The increase of $N_{ac}$ results in a larger proportion of $N_{ac}/N_{notch}$ because we set an unchanged $N_{notch}$ in this paper. The notch layer near cathode side aims to modulate the electric field of the cathode region and used as a nucleation point for the electron domain to reduce the dead zone length. However, a larger proportion of $N_{ac}/N_{notch}$ leads a larger electric field and parasitic resistance at the notch layer as well as a smaller electric field in the active region. This is why the dipole domain gets mature within a smaller distance and the domain size becomes smaller when $N_{ac}$ increases, which results in a decrease of η and $P_{RF}$. In addition, when we take the self-heating effect into consideration, the lattice temperature distribution also plays an important role on the output characteristic of the device. The temperaure profile within the device is a reflection of the distribution of the joule heating, which in turn reflects the electric field distribution. The peak temprature and heating occurs at where the field is a maximum. As shown in Figure 8, the electric field inside the notch layer enhances at larger $N_{ac}/N_{notch}$, thus the lattice temperature peak gradually transfers from the anode side to the notch layer. The lattice temperature distributes most uniformly at the doping level of $2 \times 10^{17}$ cm$^{-3}$. At the doping level of $2.5 \times 10^{17}$ cm$^{-3}$, the lattice temperature peak completely transfers to the notch layer. This is why we get the highest η and $P_{RF}$ at the doping level of $2 \times 10^{17}$ cm$^{-3}$ and no oscillation can be obtained at the doping level higher than $2.5 \times 10^{17}$ cm$^{-3}$.

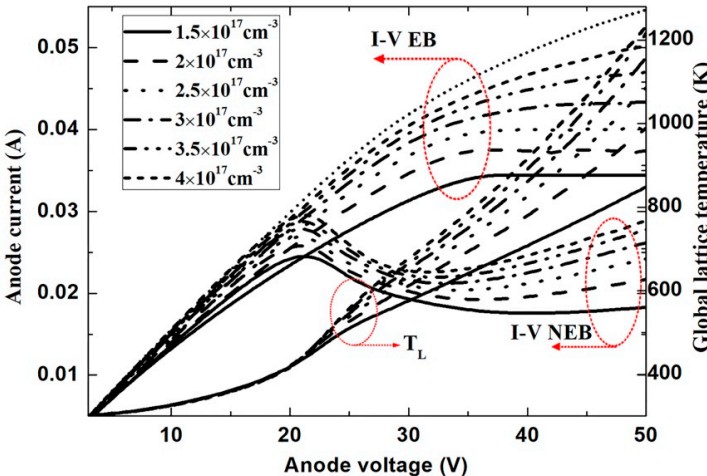

**Figure 6.** I–V characteristics of GaN Gunn diode under NEB model and EB model, and the global lattice temperature at different $N_{ac}$ under NEB model.

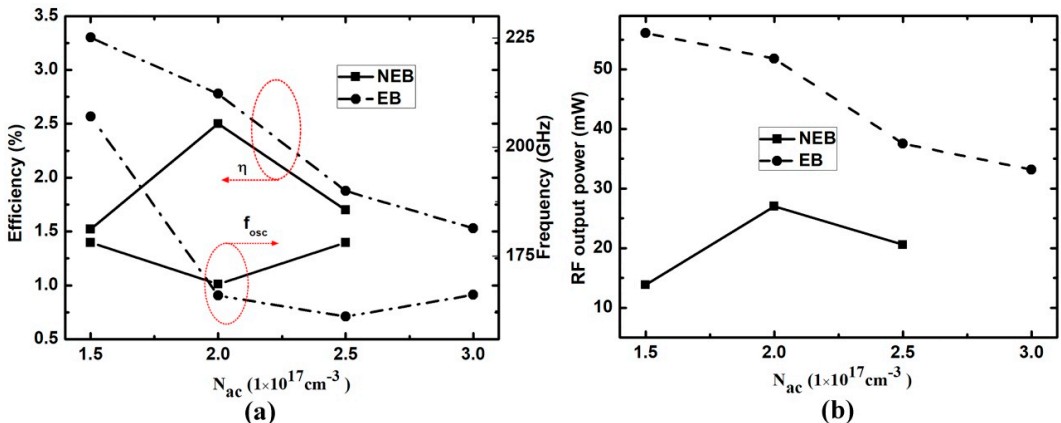

**Figure 7.** (**a**) Simulation results of the oscillation frequency $f_{osc}$ and dc-to-ac conversion efficiency $\eta$ versus the doping level $N_{ac}$ and (**b**) RF (Radio Frequency) output power $P_{RF}$ versus $N_{ac}$ under EB and NEB models.

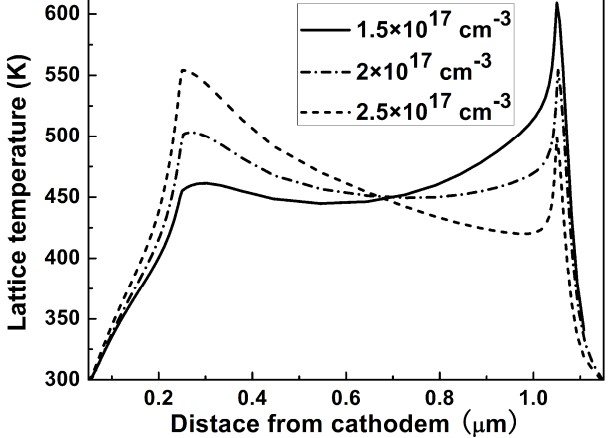

**Figure 8.** Lattice temperature profiles in the active region at different $N_{ac}$ under NEB model.

We also investigate the influence of transit region length $L_{ac}$ on output performance of GaN Gunn diode, where $L_{ac}$ changes from 0.8 to 1.4 μm at a step of 0.1 μm in this simulation. Figure 9 shows the influence of $L_{ac}$ on dc-to-ac conversion efficiency and oscillation frequency under both EB and NEB models. The dc-to-ac efficiency $\eta$ and the oscillation frequency $f_{osc}$ decrease as $L_{ac}$ increases under both

EB and NEB models, as illustrated in Figure 9. It is noteworthy that the oscillation frequency obtained under NEB model becomes larger than that obtained under EB model when $L_{ac}$ is larger than 0.9 μm. We can explain the phenomenon by using extracted electric field profiles of 1.2-μm-long device under EB and NEB model, as shown in Figure 10. Figure 10a shows that the Gunn diode operates at dipole domain mode under EB model. Figure 10b indicates that the device operates at accumulation mode as well as the dead zone is enlarged obviously when we take the self-heating effect into consideration under NEB model. The lattice temperature, particularly the temperature near the anode side, increases with $L_{ac}$. The dipole domain will become not able to sustain at higher lattice temperature, as shown in Figure 10b. Based on [9,30], in order to allow the dipole domains forming in the Gunn diode, design criteria must be met: $(N_{ac} \times L_{dc}) > (N \times L)_o \equiv 3 \times \varepsilon \times v_{peak}/q\mu_{NDR}$ ($\varepsilon$: dielectric constant; $\mu_{NDR}$: the peak negative differential mobility). The numerical value is for GaN. Therefore, most references believe that the dipole domains prefer to form at the longer transit region with higher doping level, which may only hold in ideal situation without considering the joule heating. In our simulation, we draw an absolutely opposite conclusion: the dipole domain modes easily degenerate into accumulation modes due to the self-heating effect in the longer channel. A stable oscillation hardly generates under NEB model but there is still oscillation generating under EB model when $L_{ac}$ increases to 1.4 μm.

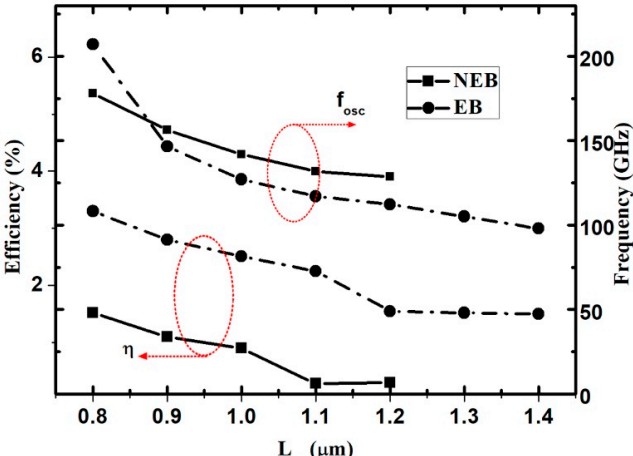

**Figure 9.** Simulation results of the oscillation frequency $f_{osc}$ and dc-to-ac conversion efficiency $\eta$ versus the active region length $L_{ac}$ under EB and NEB models.

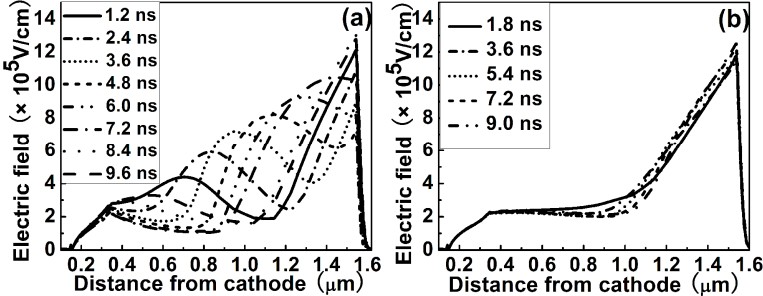

**Figure 10.** Electric field profiles of the diode derived inside one oscillation circle at the same time steps, when $L_{ac}$ = 1.2 μm, under (**a**) EB and (**b**) NEB models.

If the thermal generation is suppressed in the long transit region diode, non-unique dipole domains would appear particularly at higher doping level of $N_{ac}$. By using EB model we have found two dipole domains formed in the diode with $L_{ac}$ of 1.4 μm and $N_{ac}$ of $3 \times 10^{17}$ cm$^{-3}$, as illustrated in Figure 11a,b. The dual-dipole-domain phenomenon is attributed to two reasons. Firstly, dipole domain easily generates in the device which has higher $N_{ac} \times L_{ac}$. Secondly, the increase of the $N_{ac}/N_{notch}$ enhances the fluctuation of the electric field in the transit region, which promotes the formation of two

dipole domains inside one oscillation circle. However, the domain near the notch layer differs from the traditional dipole domain. Normally, the dipole domain consists of an excess electrons layer and a depleted electrons layer, but we cannot find an obvious excess electrons layer near the cathode side, as shown in Figure 11a. In order to demonstrate it is a dipole domain we extract electron concentration and electron velocity profiles at 12.6 ns, as shown in Figure 12a. Theoretically, the excess electrons part of the dipole domain forms as the trailing electrons behind the dipole arriving with a higher velocity. By the same token, the region of depleted electrons also grows because electrons ahead of the dipole leave at a higher velocity [30]. The correspondence between the electron concentration and electron velocity shown in Figure 12a excellently supports this opinion. The electrons inside the notch layer experience a velocity overshoot, as illustration in the region 1 of the Figure 12a. So the electrons of the excess electrons layer comes from the notch layer initially. One on hand, the electron concentration of notch layer is greatly lower than that of the active channel; on the other hand, the excess electrons layer part comes into being immediately inside the notch layer. Therefore, it results in the formation of an obscure excess electrons layer. At the same time, the electrons in region 2 speed up, ahead of which formed a depleted electrons layer. Due to the dual-dipole-domain phenomenon inside one oscillation circle, the harmonic frequency is greatly enhanced, as shown in Figure 12b where the frequency spectrum is extracted by means of the fast Fourier transformation (FFT) algorithm. The excellent frequency multiply would find a promising application in terahertz technology.

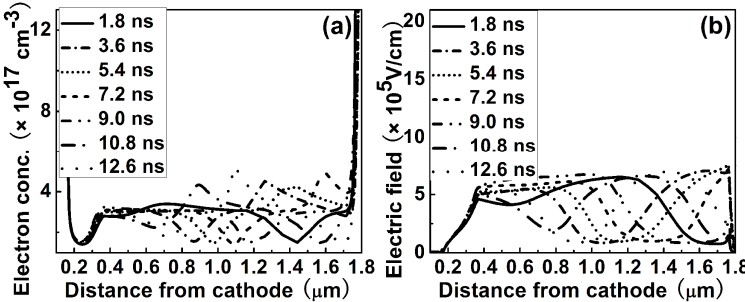

**Figure 11.** Electron concentration (**a**) and electric field (**b**) profiles derived inside one oscillation circle at the same time step, when $L_{ac} = 1.4$ μm and $N_{ac} = 3 \times 10^{17}$ cm$^{-3}$ under EB model.

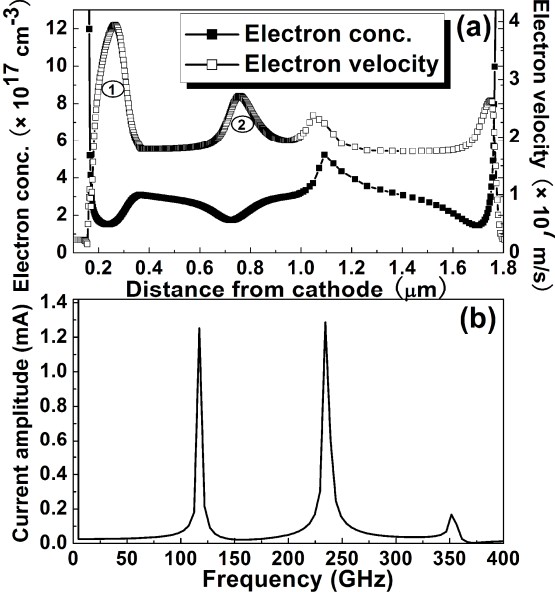

**Figure 12.** (**a**) The relationship of the electron velocity and electron concentration derived inside on oscillation circle at 12.6 ns, when $L_{ac} = 1.4$ μm, $N_{ac} = 3 \times 10^{17}$ cm$^{-3}$; (**b**) and its frequency spectrum diagrams.

## 4. Conclusion

In this paper, we present a numerical analysis on self-heating effect in GaN vertical $n^+$-$n^-$-$n$-$n^+$ Gunn diode based on Silvaco-Atlas, by emphasizing the microscopic analysis of the electron domain mode. In order to better understand the thermal and electrical properties of GaN Gunn diode, two models—the non-isothermal energy balance model (NEB) and energy balance model (EB)—are solved and analyzed alternately in the implementation of each sample. Our simulations have demonstrated that self-heating effect degrade the electron domain model and output characteristics of GaN Gunn diode. There is an obvious decrease of the saturation current in the direct I–V characteristics of each sample under NEB model, and peak lattice temperature in the notch region and reaches more than 1015 K at the operation voltage of 60 V. In order to avoid the serious heat generation, pulsed current is suggested instead of direct current. We have also explored the influence of the various parameters, viz. operation bias, the doping level and length of the active region on the heating generation and distribution. The increase of these parameters increases the inelastic phonon scattering processes of the electron, and further enhances the heat generation of the lattice, which weakens the NDR effect and degrades the electron domain as well as the output characteristics. In addition, with self-heating effect taken into consideration, the normal operation range of device is greatly limited, as compared with the ideal situation. We propose the transition domain for the first time, which is in between the dipole domain and accumulation layer, and stands for the degradation of the electron domain. A proper $N_{ac}/N_{notch}$ not only makes the notch layer a nucleate layer but also provides a more uniform temperature distribution, which results in higher output characteristics of the device. As demonstrated in our simulation, best performance is obtained in the sample with a doping level of $2 \times 10^{17}$ cm$^{-3}$. Dual-domain phenomenon is formed under EB model in the device with longer active region length and higher $N_{ac}/N_{notch}$, which enhances the harmonic frequency. However, we have not found dual-domain phenomenon under NEB model. The simulation of the structure parameters makes it possible to optimize the device and reduce the self-heating effect. Other engineering approaches to alleviate the heating like using an effective heat-sinking are our future work.

**Author Contributions:** Conceptualization, Y.W. and J.A.; methodology, Y.W.; software, Y.W.; validation, Y.W., J.A. and S.L.; formal analysis, Y.W.; investigation, W.Y.; resources, Y.H.; data curation, Y.W.; writing—original draft preparation, Y.W.; writing—review and editing, S.L., Y.H.; visualization, J.A.; supervision, J.A., Y.H.; project administration, J.A.; funding acquisition, J.A., Y.H.

**Funding:** This work was partially supported by the National Key Research and Development Program (Grant No. 2017 YFB043000) and the National Natural Science Foundation of China (Grant No. 61274092).

**Acknowledgments:** The author Ying Wang thanks the colleagues and her teachers who help her in her research work.

**Conflicts of Interest:** The authors declare no conflict of interest.

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
