# Peer review of "Thermal Modeling of the GaN-based Gunn Diode at Terahertz Frequencies"

_applsci, doi:10.3390/app9010075_

Round 1
Reviewer 1 Report
In the manuscript, the authors theoretically study GaN-based Gunn diode taking into account thermal effects, self-heating, on the diode performance. I find results interesting and worth of publication providing the manuscript revision outlined below.
1. The manuscript needs an addition figure (preferably the very first one) that should schematic of the considered structure, interfaces, and what processes are accounted for.
2. In turn, the manuscript contains too many plots with similar information. On top of it, the figures are not explained in a very detail. I recommend the authors to remove the least informative plots and consolidate the remaining plots in several figures with multiple panels based on similarity of information. The captions need to be extended. Ideally, captions need to be sufficient to understand plots and processes on them without reading the main part of the manuscript.
3. In line 83, the authors mention “In order to simplify the algorithm, the heat transport from the metal contact to external surroundings is not included in our simulations.” It will be good if the authors provide a discussion of possible changes in a model advances model if the thermal transport from metal is taken into account.
4. There is an issue with the introduction of abbreviations: sometimes it is not done at all and sometimes it is somewhere in the middle of the text (not upon the first use). For instance “NDR” is written in line 24 and fully spelled out only in line 30. It is also spelled out in line 85 which is unnecessary. In line 46, the term “HEMT” also needs to be explained. The same is about “FFT” in line 244.
5. In lines 66 and 68-69, the authors mention “Atlas”; in line 71 they write “ATLAS”, and only in line 74 the authors give the reference on the software. One needs to make the reference earlier in the text, preferably line 66.
6. The very first equation (line 75) has number (4) and should be (1) if overall the number is in use.
7. In line 86, the authors write “both particles”. They need to explicitly mention what particles they are talking about.
8. In line 35: ”may not stand without taking the self-heating”. It seems there is something wrong with the phase and probably “without” should be replaced by “when”. Alternative corrections may also work.
9. In the abstract, in the phrase “the negative effect of self-heating effect”, the second word “effect” can be removed
10. In line 207 “Figure 9 shows under both EB and NEB models.” However, it is not said what is shown and the sentence needs to be extended by giving more details about the situation.
11. Misprints and style: line 50 “sates” -> “states”; line 51 “we analysis” -> “we analyze”; in line 123 “field is a maximum” -> “field has a maximum”; “As a conclusion” -> “Consequently”; line 174 “temperature” -> “temperature”; in line 236 “excellently supports this opinion” -> “excellently supports this prediction” (or “statement”, or “hypothesis”); in line 250 “(a) EB model and (b) NEB model” -> “(a) EB and (b) NEB models”
Author Response
1. The manuscript needs an addition figure (preferably the very first one) that should schematic of the considered structure, interfaces, and what processes are accounted for.
Thanks very much for your suggestion. As far as we are concerned, a schematic of the device is necessary, which is added in the revised manuscript as Fig. 1. And the order of all the figures is readjusted.
Figure 1 Schematic structures of n+-n--n-n+ structural GaN Gunn didoes.
2. In turn, the manuscript contains too many plots with similar information. On top of it, the figures are not explained in a very detail. I recommend the authors to remove the least informative plots and consolidate the remaining plots in several figures with multiple panels based on similarity of information. The captions need to be extended. Ideally, captions need to be sufficient to understand plots and processes on them without reading the main part of the manuscript.
Thanks for your suggestion. Based on your opinions, we have cut out one of these figures (Fig. 7 in the original manuscript) and extended the captions of all figures. Actually as shown in the revised manuscript, Fig. 3, Fig. 7 and Fig. 9 almost shows the same content. However, if all the curves of these figures are drawn in one figure, the curves will seem a bit confusing and become hard to describe and discuss. And Fig. 3 and 7 contains two figures (a) and (b). Therefore, we do not merge these figures in the revised manuscript.
3. In line 83, the authors mention “In order to simplify the algorithm, the heat transport from the metal contact to external surroundings is not included in our simulations.” It will be good if the authors provide a discussion of possible changes in a model advances model if the thermal transport from metal is taken into account.
Thanks for your suggestion. There are mainly two reasons for choosing such thermal boundary conditions. Firstly, in order to simplify the algorithm, most of the literatures on the simulation of the thermal effect of the semiconductor device have adopted this simplified method without taking account the heat transport form the metal contact to external surroundings, as described in Ref. 1-3. The examples on thermal effect shown offered by Silvaco officially also adopt the same method. Secondly, it will increase the calculation amount and result in convergence problems, if heat transport from the metal to external surroundings is included in the simulation code. We have not been able to solve the non-convergence problem. Therefore, in order to simplify the algorithm, the heat transport from the metal contact to external surroundings is not included in our simulations. It will introduce some errors in the results as compared with the practice devices. However, this paper is aimed at investigating the mechanism of electron domains and self-heating effect in a Gunn diode rather than the temperature values or output performance of a NDR oscillator. Appropriate modifications have been given in the revised manuscript.
Reference
1. A. K. Sahoo, N. K. Subramani, J. C. Nallatamby, et al. Microwave Integrated Circuits Conference. IEEE, 2016.
2. S. García, I Íñiguez-de-la-Torre, J Mateos, T González and S Pérez. Semiconductor Science and Technology 31. 0650056 (2016).
3. J. P. Jones, E. Heller, D. Dorsey, S. Graham, Microelectronics Reliability. 55, 2634, (2015).
4. There is an issue with the introduction of abbreviations: sometimes it is not done at all and sometimes it is somewhere in the middle of the text (not upon the first use). For instance “NDR” is written in line 24 and fully spelled out only in line 30. It is also spelled out in line 85 which is unnecessary. In line 46, the term “HEMT” also needs to be explained. The same is about “FFT” in line 244.
Thanks for your reminding. I have given explanation of these abbreviations upon the first use in the revised manuscript. NDR: negative differential resistance; HEMT: high electron mobility transistor; FFT: Fast Fourier Transformation.
5. In lines 66 and 68-69, the authors mention “Atlas”; in line 71 they write “ATLAS”, and only in line 74 the authors give the reference on the software. One needs to make the reference earlier in the text, preferably line 66.
Thanks for your question. I have changed all “ATLAS” as “Atlas” and adjusted the order of the Ref. 22 and Ref. 23, as shown in the Reference part.
6. The very first equation (line 75) has number (4) and should be (1) if overall the number is in use.
Thanks very much for your reminding, I have revised the number in the revised manuscript.
7. In line 86, the authors write “both particles”. They need to explicitly mention what particles they are talking about.
In Sivaco Atlas, the Energy balance model is selected by specifying the HCTE.EL parameters in the MODELS statement, which will enable the calculation of the energy balance equations for the electron. To enable the non-isothermal energy balance model, the heat flow equations should also be solved on the basis of energy balance equations, by specifying the “LAT.TEMP” parameters in the MODELS statement. Therefore, the “both particle continuity equations” means the “energy balance equations” and the “heat flow equations”. In order to make it easier to understand, we have revised this sentence as: “We have used a non-isothermal energy balance (NEB) model by specifying both HCTE.EL and LAT.TEMP parameters in the MODELS statement to solve both energy balance equations and heat flow equations.”
8. In line 35: “may not stand without taking the self-heating”. It seems there is something wrong with the phase and probably “without” should be replaced by “when”. Alternative corrections may also work.
Thanks very much for your reminding, I have made the corrections on the revised manuscript.
9. In the abstract, in the phrase “the negative effect of self-heating effect”, the second word “effect” can be removed.
Thanks very much for your reminding. The abstract has been revised.
10. In line 207 “Figure 9 shows under both EB and NEB models.” However, it is not said what is shown and the sentence needs to be extended by giving more details about the situation.
Thanks very much for your reminding. The sentence of “Figure 9 shows under both EB and NEB models.” has been revised as “Figure 9 shows the influence of Lac on dc-to-ac conversion efficiency and oscillation frequency under both EB and NEB models.”
11. Misprints and style: line 50 “sates”-> “states”; line 51 “we analysis” -> “we analyze”; in line 123 “field is a maximum” -> “field has a maximum”; “As a conclusion” -> “Consequently”; line 174 “temperature” -> “temperature”; in line 236 “excellently supports this opinion” -> “excellently supports this prediction” (or “statement”, or “hypothesis”); in line 250 “(a) EB model and (b) NEB model” -> “(a) EB and (b) NEB models”
Thanks for all of your suggestions. I have revised all the mistakes based on your suggestions, as shown in the revised paper.

Reviewer 2 Report
The authors discribe the self-heating effect on the Gunn diode based on GaN, which is important for THz application. Self-heating is a common phenomineon in electronic devices, especially those devices with high power output. In order to correctly understand the electronic transport in the elecronic devices, self-heating is unevitable. This paper gives a good example to study this important phenomineon and is quite interesting. I thus recommend the acceptance of this paper, providing that the authours make the following revisions.
(1) It will be more readable if a device configuration is provided.
(2) In Fig. 1 and other figures where electric current is shown, it will be better to use current density (i.e. A/cm2 ) instead of A.
(3) Sine heating finally depends on the output power, can you also show the power dependent temperature increase effect in the devices ?
Author Response
1. It will be more readable is a device configuration is provided
Thanks very much for your suggestion. Based on your opinion the schematic of the device is added in the revised manuscript as Fig. 1. And the order of all the figures is readjusted.
Figure 1 Schematic structures of n+-n--n-n+ structural GaN Gunn didoes.
(2) In Fig. 1 and other figures where electric current is shown, it will be better to use current density (i.e. A/cm2) instead of A.
Thanks very much for your suggestion. We have supplemented the parameters of the devices simulated in this paper. The width of all the diodes is a default of 1 μm in the two-dimensional ATLAS simulator. And we set the device height as 5 μm. As all the diodes have the same cross-sectional area, we think it may be ok to describe the current by “A” instead of “A/cm2”. As we have also given the values of efficiency, the RF output power can be estimated faster based on the current value.
(3) Since heating finally depends on the output power, can you also show the power dependent temperature increase effect in the devices?
Thanks very much for your suggestion. We have given the relation curves of RF output power PRF verse Nac and Lac in Fig. 3 and Fig. 7.
Figure 3. (a)Simulation results of the oscillaiton frequency fosc and dc-to-ac conversion efficiency h versus the anode voltage Vdc and (b) RF output power PRF versus Vdc under EB and NEB models.
Figure 7. (a)Simulation results of the oscillaiton frequency fosc and dc-to-ac conversion efficiency h versus the doping level Nac and (b) RF output power PRF versus Nac under EB and NEB models.

Reviewer 3 Report
My comments are as follows:
- Where are the studied structures? Where are the layers? The studied structures should be given as figures.
- The authors should provide further arguments to demonstrate that not considering the heat loss from the metal contacts to surroundings in the simulations is a valid approach.
- The authors claim they are the first ones to propose a transition domain for the first time (line 273). This claim should be also present in the abstract and in the manuscript body.
- What do the authors mean by stating "dipole domain mode prefers to come into being in longer channel"? Explain in a more clear way.
- There are paragraphs which are very long in the manuscript. It is very difficult to follow for the reader. Break them up plausibly.
- Fig. 5 has two references to I-V NEB, but no reference to I-V EB. One of the references should be to I-V EB (top one). Fix accordingly.
- The sentence between lines 87-89 should include a reference to literature source.
- The abstract should be modified in a way that it summarize the most important findings in the study.
- Line 38-39. What are the "limitations"? Be specific.
- First paragraph of Introduction. What is NDR? Give the extension.
- Line 59-63. The sentence is very long and hard to follow. Break it up into manageable pieces.
- The first three sentences starting at line 78 should include references to literature sources. Where did you get these equations and constants?
- What is LAT.TEMP and HCTE.EL? What is MODELS statement? Either remove them or explain them.
- What is HEMT? Give the extension.
Author Response
1. Where are the studied structures? Where are the layers? The studied structures should be given as figures.
Thanks very much for your suggestion. The schematic of the device is added in the revised manuscript as Fig. 1.
Figure 1 Schematic structures of n+-n--n-n+ structural GaN Gunn didoes.
2. The authors should provide further arguments to demonstrate that not considering the heat loss from the metal contacts to surroundings in the simulations is a valid approach.
Thanks for your suggestion. There are mainly two reasons for choosing such thermal boundary conditions. Firstly, in order to simplify the algorithm most of the literatures on the simulation of the thermal effect of GaN diode have adopted this simplified method without taking account the heat transport form the metal contact to external surroundings, as described in Ref. 1-3. The examples on thermal effect shown in Silvaco official website also adopt the same method. Secondly, it will increase the calculation amount and result in convergence problems, if heat transport from the metal to external surroundings is included in the simulation code. We have not been able to solve the non-convergence problem. Therefore, in order to simplify the algorithm, the heat transport from the metal contact to external surroundings is not included in our simulations. It will introduce some errors in the results as compared with the practice devices. This paper is aimed at investigating the mechanism of electron domains and self-heating effect in a Gunn diode rather than the temperature values or output performance of a NDR oscillator. Appropriate modifications have been given in the revised manuscript.
Reference
1. A. K. Sahoo, N. K. Subramani, J. C. Nallatamby, et al. Microwave Integrated Circuits Conference. IEEE, 2016.
2. S. García, I Íñiguez-de-la-Torre, J Mateos, T González and S Pérez. Semiconductor Science and Technology 31. 0650056 (2016).
3. J. P. Jones, E. Heller, D. Dorsey, S. Graham, Microelectronics Reliability. 55, 2634, (2015).
3. The authors claim they are the first ones to propose a transition domain for the first time (line 273). This claim should be also present in the abstract and in the manuscript body.
Thanks very much for your suggestion. A newly revised abstract has been given in the revised manuscript, which contains the “transition domain”. It also has been emphasized in the manuscript body.
Abstract: In this paper, a comprehensive evaluation of thermal behavior of the GaN vertical n+-n--n-n+ Gunn diode have been carried out through simulation method. We explore the complex effects of various parameters on the device thermal performance through a microscopic analysis of electron movements. These parameters include operation bias, doping level and length of the active region. The increase of these parameters aggravates the self-heating effect and degrades the electron domains, which therefore reduces the overall performance output of the diode. However, appropriate increase of the doping level of active region makes the lattice heat distribute more uniformly and improves the device performance. For the first time, we propose the transition domain, which is in between the dipole domain and accumulation layer, and stands for the degradation of the electron domain. We have also demonstrated that dual domains occur in the device with longer active region length and higher Nac/Nnotch under EB model, which enhances the harmonics component. Electric and thermal behaviors analysis of GaN vertical Gunn diode makes it possible to optimize the device.
4. What do the authors mean by stating "dipole domain mode prefers to come into being in longer channel"? Explain in a more clear way.
In order to allow the dipole domains forming in the Gunn diode, design criteria must be met: (NA×LA)>(N×L)o≡3×ε×vpeak/qμNDR≈2-3×1013cm-2 (NA: the doping level of the transit region; LA: the length of the transit region; ε: dielectric constant; μNDR: the peak negative differential mobility)1,2. The numerical value is for GaN. Therefore the dipole domains prefer to form at the longer transit region with higher doping level. Similar results are also given in the reference 3.
Reference
1. J. T. Lü, and J. C. Cao, Semicond. Sci. Technol., 19, 451 (2004)
2. S. M. Sze and K. K. Ng Physics of Semiconductor Devices, 3rd ed. (Wiley. Hoboken. 2007).
3. Y. Wang, L. A. Yang, W. Mao, S. Long, Y. Hao, IEEE Transactions on Electron Devices, 60, 1600 (2013).
5. There are paragraphs which are very long in the manuscript. It is very difficult to follow for the reader. Break them up plausibly.
Thanks very much for your suggestion. We have tried our best to revise the paper, and make it readable.
6. Fig. 5 has two references to I-V NEB, but no reference to I-V EB. One of the references should be to I-V EB (top one). Fix accordingly.
Thanks very much for your reminding. We have modified this figure in the revised manuscript.
Figure 6. I-V characteristics of GaN Gunn diode under NEB model and EB model, and the global lattice temperature at different Nac under NEB model.
7. The sentence between lines 87-89 should include a reference to literature source.
Thanks for your suggestion. The reference has been added in the revised manuscript as Ref. 22.
22. Atlas, Device Simulator. “Atlas User’ Manual,” Silvaco International Software, Santa Clara, CA, USA, (2010).
8. The abstract should be modified in a way that it summarizes the most important findings in the study.
Thanks very much for your suggestion. The following is the revised abstract.
Abstract: In this paper, a comprehensive evaluation of thermal behavior of the GaN vertical n+-n--n-n+ Gunn diode have been carried out through simulation method. We explore the complex effects of various parameters on the device thermal performance through a microscopic analysis of electron movements. These parameters include operation bias, doping level and length of the active region. The increase of these parameters aggravates the self-heating effect and degrades the electron domains, which therefore reduces the overall performance output of the diode. However, appropriate increase of the doping level of active region makes the lattice heat distribute more uniformly and improves the device performance. For the first time, we propose the transition domain, which is in between the dipole domain and accumulation layer, and stands for the degradation of the electron domain. We have also demonstrated that dual domains occur in the device with longer active region length and higher Nac/Nnotch under EB model, which enhances the harmonics component. Electric and thermal behaviors analysis of GaN vertical Gunn diode makes it possible to optimize the device.
9. Line 38-39. What are the "limitations"? Be specific.
Firstly, one of the limitations is there has been no any experimental report on the rf oscillation of GaN based bulk or HEMT-liked planar Gunn diode due to the technological bottleneck on GaN epitaxial growth and device fabrication and the serious self-heating problem, as which may trigger the weakening of the NDR effect and even the permanent failure of diode. Therefore, simulation method is an easier and better way to study the self-heating effect. Secondly, the self-heating effect deteriorates the output characteristics of the Gunn diode by affecting the motion of the electrons. It is difficult to provide a microscopic analysis on electron movements (especially the electron domain movements) by experimental method. Thirdly, the device fabrication and measurement is often very difficult because of long duration, and high expense. Comprehensive theory research will increase the success rate of experimental research. Explicit explanation on “limitations” has been given in revised manuscript: “There are great limitations to study the device thermal performance on the experimental method. Firstly, as we mention above, the fabrication and measurement of GaN Gunn diode are still very difficult because of bottlenecks on GaN epitaxial growth and device fabrication and the serious self-heating problem. Secondly, the self-heating effect deteriorates the output characteristics of the Gunn diode by affecting the motion of the electrons. However, it is difficult to provide a microscopic analysis on electron movements (especially the electron domain movements) by experimental method. Therefore, we choose the physical-based modeling and simulation method to achieve accurate device thermal management, which would guide the experiment research in turn.”
10. First paragraph of Introduction. What is NDR? Give the extension.
NDR is the abbreviation of “Negative Differential Resistance”, which has been explained in the revised manuscript.
11. Line 59-63. The sentence is very long and hard to follow. Break it up into manageable pieces.
Thanks for your suggestion. The longer sentence “Compared with Drift-Diffusion (DD) model, EB model is more accurate for spatiotemporal domain and chaos in the short channel because it includes non-localized carrier transport phenomena, such as the velocity overshoot and the non-local impact ionization which are neglected by the conventional DD model, so it is more suitable for the transient simulation of terahertz devices and is employed in this paper.” has been broken up into “Compared with Drift-Diffusion (DD) model, EB model is more accurate for spatiotemporal domain and chaos in the short channel. This is because the EB model includes non-localized carrier transport phenomena, such as the velocity overshoot and the non-local impact ionization which are neglected by the conventional DD model. Therefore, the EB model is more suitable for the transient simulation of terahertz devices and is employed in this paper.” in the revised manuscript.
12. The first three sentences starting at line 78 should include references to literature sources. Where did you get these equations and constants?
The main research literature that we referenced to achieve these equations and constants has been illustrated in the revised manuscript as Ref. 23.
23. B. E. Fouz, L. F. Eastman, U. V. Bhapkar, and M. S. Shur, Appl. Phys. Lett., 70, 2849 (1997).
13. What is LAT.TEMP and HCTE.EL? What is MODELS statement? Either remove them or explain them.
Thanks for your questions. LAT.TEMP and HCTE.EL, are command statements in the Silvaco Atlas. HCTE.EL is specified to invoking equations for energy balance model; LAT.TEMP is specified to invoking the heat flow equations. MODELS statement is the important part in the device-simulation code, where the physical models of the device is specified. LAT.TEMP and HCTE.EL are specified in the MODELS statement to call different equations that we need. A brief explanation of LAT.TEMP , HCTE.EL, and MODELS statement have been given in the revised manuscript.
14. What is HEMT? Give the extension.
Thanks for your questions. I have given explanations to all the abbreviations in the revised manuscript. HEMT: High Electron Mobility Transistor

Round 2
Reviewer 1 Report
All my comments and suggestions have been taking into account. The suggestions are reflected in the manuscript, it has been changed accordingly and significantly improved. I recommend manuscript publication.
Reviewer 3 Report
Thanks for the revision of the manuscript based on the suggestions.